# GATED RECURRENT NEURAL NETWORKS DISCOVER ATTENTION

## ABSTRACT

Recent architectural developments have enabled recurrent neural networks (RNNs) to reach and even surpass the performance of Transformers on certain sequence modeling tasks. These modern RNNs feature a prominent design pattern: linear recurrent layers interconnected by feedforward paths with multiplicative gating. Here, we show how RNNs equipped with these two design elements can exactly implement (linear) self-attention, the main building block of Transformers. By reverse-engineering a set of trained RNNs, we find that gradient descent in practice discovers our construction. In particular, we examine RNNs trained to solve simple in-context learning tasks on which Transformers are known to excel and find that gradient descent instills in our RNNs the same attention-based in-context learning algorithm used by Transformers. Our findings highlight the importance of multiplicative interactions in neural networks and suggest that certain RNNs might be unexpectedly implementing attention under the hood.

## 1 INTRODUCTION

Attention-based neural networks, most notably Transformers (Vaswani et al., 2017), have rapidly become the state-of-the-art deep learning architecture, replacing traditional models such as multi-layer perceptrons, convolutional neural networks, and recurrent neural networks (RNNs). This is particularly true in the realm of sequence modeling, where once-dominating RNNs such as the long short-term memory (LSTM; Hochreiter & Schmidhuber, 1997) model and the related gated recurrent unit (GRU; Cho et al., 2014) have been mostly replaced by Transformers.

Nevertheless, RNNs remain actively researched for various reasons, such as their value as models in neuroscience (Dayan & Abbott, 2001), or simply out of genuine interest in their rich properties as a dynamical system and unconventional computer (Jaeger et al., 2023). Perhaps most importantly for applications, RNNs are able to perform inference for arbitrarily long sequences at a constant memory cost, unlike models based on conventional softmax-attention layers (Bahdanau et al., 2015). This ongoing research has led to a wave of recent developments. On the one hand, new deep linear RNN architectures (Gu et al., 2022; Orvieto et al., 2023b) have been shown to significantly outperform Transformers on challenging long-sequence tasks (e.g., Tay et al., 2020). On the other hand, efficient linearized attention models have been developed, whose forward pass can be executed in an RNN-like fashion at a constant inference memory cost (Tsai et al., 2019; Katharopoulos et al., 2020; Choromanski et al., 2021; Schlag et al., 2021; Fu et al., 2023).

We present a unifying perspective on these two seemingly unrelated lines of work by providing a set of parameters under which gated RNNs become equivalent to any linearized self-attention, without requiring infinite number of neurons or invoking a universality argument. Crucially, our construction makes use of gated linear units (GLUs; Dauphin et al., 2017), which are ostensibly featured in recent deep linear RNN models. Turning to LSTMs and GRUs, which also include multiplicative gating interactions, we find somewhat surprisingly that our results extend only to LSTMs. Moreover, the LSTM construction we provide requires a very specific configuration, which hints that the inductive bias towards attention-compatible configurations might be weaker for this architecture than for deep gated linear RNNs.

We then demonstrate that GLU-equipped RNNs, but not LSTMs and GRUs, can effectively implement our construction once trained, thus behaving as attention layers. Moreover, we find that such GLU-equipped RNNs trained to solve linear regression tasks acquire an attention-based in-context

learning algorithm. Incidentally, it has been shown that the very same algorithm is typically used by Transformers trained on this problem class (von Oswald et al., 2023; Mahankali et al., 2023; Ahn et al., 2023; Zhang et al., 2023). Our results thus challenge the standard view of RNNs and Transformers as two mutually exclusive model classes and suggest that, through learning, RNNs with multiplicative interactions may end up encoding attention-based algorithms disguised in their weights.

## 2 BACKGROUND

### 2.1 LINEAR SELF-ATTENTION

We study causally-masked linear self-attention layers that process input sequences $(x_t)_t$ with $x_t \in \mathbb{R}^d$ as follows:

$$y_t = \left( \sum_{t' \leq t} (W_V x_{t'})(W_K x_{t'})^\top \right) (W_Q x_t) \tag{1}$$

In the previous equation, $W_V \in \mathbb{R}^{d \times d}$ is the value matrix, $W_K \in \mathbb{R}^{d \times d}$ the key matrix and $W_Q \in \mathbb{R}^{d \times d}$ the query matrix. We use square matrices throughout the paper for simplicity, but our findings extend to rectangular ones. As usually done, we call $v_t := W_V x_t$, $k_t := W_K x_t$ and $q_t := W_Q x_t$ the values, keys and queries. The output vector $y_t$ has the same dimension as the input, that is $d$. Such linear self-attention layers can be understood as a linearized version of the softmax attention mechanism (Bahdanau et al., 2015) in use within Transformers (Vaswani et al., 2017). Yet, they operate in a very different regime than softmax layers, which have unbounded memory. Attention layers commonly combine different attention heads; we focus on a single one here for simplicity.

In a linear self-attention layer, information about the past is stored in an effective weight matrix $W_t^{\text{ff}} := \sum_{t'} v_{t'} k_{t'}^\top$ that will later be used to process the current query $q_t$ through $y_t = W_t^{\text{ff}} q_t$. At every timestep, $W_t^{\text{ff}}$ is updated through the rule $W_t^{\text{ff}} = W_{t-1}^{\text{ff}} + v_t k_t^\top$, which is reminiscent of Hebbian learning (Schmidhuber, 1992; Schlag et al., 2021) and leads to faster inference time (Katharopoulos et al., 2020; Choromanski et al., 2021; Shen et al., 2021; Peng et al., 2021) than softmax self-attention.

### 2.2 GATED RECURRENT NEURAL NETWORKS

In this paper, we focus our analysis on a simplified class of gated diagonal linear recurrent neural networks. They implement bilinear input $g^{\text{in}}$ and output gating $g^{\text{out}}$ that multiplies a linear transformation $W_{\text{x}}^{\text{in/out}} x_t$ of the input with a linear gate $W_{\text{m}}^{\text{in/out}} x_t$: $g^{\text{in/out}}(x_t) = (W_{\text{m}}^{\text{in/out}} x_t) \odot (W_{\text{x}}^{\text{in/out}} x_t)$. Here, $\odot$ is the element-wise product. The class of gated networks we consider satisfies

$$h_{t+1} = \lambda \odot h_t + g^{\text{in}}(x_t), \quad y_t = D g^{\text{out}}(h_t). \tag{2}$$

In the previous equation, $\lambda$ is a real vector, $x_t$ is the input to the recurrent layer, $h_t$ the hidden state, and $D$ a linear readout. This simplified class makes connecting to attention easier while employing similar computational mechanisms as standard gated RNNs architectures.

This class is tightly linked to recent deep linear RNN architectures and shares most of its computational mechanisms with them. While linear diagonal recurrence might be seen as a very strong inductive bias, many of the recent powerful deep linear RNN models adopt a similar bias (Gupta et al., 2022; Smith et al., 2023), and it has been shown to facilitate gradient-based learning (Orvieto et al., 2023b; Zucchet et al., 2023b). Those architectures use complex-valued hidden states in the recurrence; we only use its real part here. Some of those works employ a GLU (Dauphin et al., 2017) after each recurrent layer, with $\text{GLU}(x) = \sigma(W_{\text{m}} x_t) \odot W_{\text{x}} x_t$ with $\sigma$ the sigmoid function. The gating mechanism we consider can thus be interpreted as a linearized GLU. Finally, we can recover (2) by stacking two layers: the GLU in the first layer acts as our input gating, and the one in the second as output gating. We include a more detailed comparison in Appendix A. In the rest of the paper, we will use the LRU layer (Orvieto et al., 2023b) as the representative of the deep linear RNN architectures because of its proximity with (2).

LSTMs can operate in the regime of Equation 2, but this requires more adaptation. First, the recurrent processing of these units is nonlinear and is more involved than a simple matrix multiplication

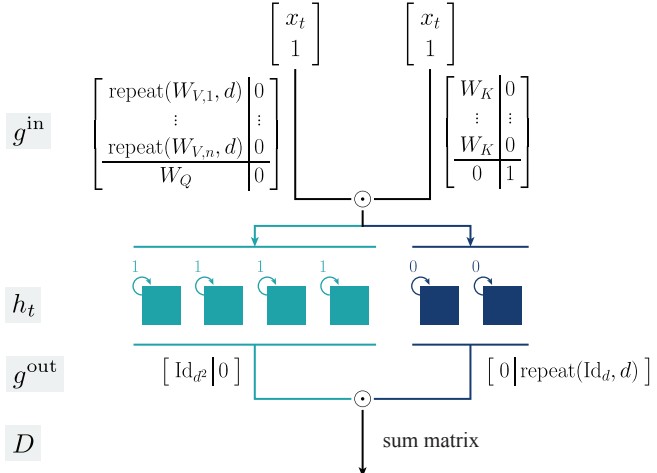

Figure 1: An example of a diagonal linear gated recurrent neural network that implements the same function as a linear self-attention layer with parameters $(W_V, W_K, W_Q)$ and input dimension $d$, as described in Section 3. Inputs are processed from top to the bottom. We do not use biases so we append 1 to the input vector $x_t$ to be able to send queries to the recurrent neurons. We use $\text{repeat}(A, n)$ to denote that the matrix $A$ is repeated $n$ times on the row axis and $W_{V,i}$ is the $i$-th row of the $W_V$ matrix. The bars within the matrices separate the different kinds of inputs/outputs. Digits in matrices denote column vectors appropriately sized. The readout matrix $D$ appropriately sums the element-wise products between key-values and queries computed after the output gating $g^{\text{out}}$.

followed by a nonlinearity. Second, gating occurs in different parts of the computation and depends on additional variables. We compare in more details this architecture and the one of Equation 2 in Appendix A, showing that LSTMs can implement (2) when stacking two layers on top of each other. We additionally show that GRUs cannot do so.

## 3 THEORETICAL CONSTRUCTION

As highlighted in the previous section, our class of gated RNNs and linear self-attention have different ways of storing past information and using it to modify the feedforward processing of the current input. The previous state $h_t$ acts through a bias term $\lambda \odot h_t$ that is added to the current input $g^{\text{in}}(x_t)$ in gated RNNs, whereas the linear self-attention recurrent state $W_t^{\text{ff}}$ modifies the weights of the feedforward pathway. We reconcile these two mismatched views of neural computation in the following by showing that gated RNNs can implement linear self-attention.

In this section, we demonstrate how a gated recurrent layer followed by a linear readout as in Equation 2 can implement any linear self-attention layer through a constructive proof. In particular, our construction only requires a finite number of neurons to exactly match the desired function, therefore providing a much stronger equivalence result than more general universality of linear recurrent networks theorems (Boyd & Chua, 1985; Orvieto et al., 2023a), which hold in the limit of infinitely many recurrent neurons.

### 3.1 KEY IDEAS

Our construction comprises three main components: Firstly, the input gating $g^{\text{in}}$ is responsible for generating the element-wise products between the keys and values, as well as the queries. Then, recurrent units associated with key-values accumulate their inputs with $\lambda = 1$, whereas those receiving queries as inputs return the current value of the query, hence $\lambda = 0$. Lastly, the output gating $g^{\text{out}}$ and the final readout layer $D$ are in charge of multiplying the flattened key-value matrix with the query vector. We illustrate our construction and provide a set of weights for which the functional equivalence holds in Figure 1. Crucially, the key-values in a linear self-attention layer are the sum of

degree two polynomials of each previous input. Input gating mechanism and perfect memory units ($\lambda = 1$) are needed to replicate this behavior within a gated recurrent layer. Similarly, output gating is required to multiply key-values with the queries.

## 3.2 ON THE NUMBER OF NEURONS REQUIRED BY THE CONSTRUCTION

The construction of Figure 1 requires $d^2 + d$ hidden neurons to store all the entries of the $d \times d$ key-value matrix and of the query vector of size $d$. While this construction is arguably the most intuitive, it is not optimal in terms of number of neurons used. Knowing the exact minimal number of neurons is fundamental for understanding which solution the network learns. Therefore, we explain in the following how to modify our construction accordingly. We leverage two additional insights: First, any combination of key and query matrices for which $(W_K^\top W_Q)$ is fixed leads to the same function in the linear self-attention layer. We can thus assume that the key and value matrices are equal, as taking the key matrix to be equal to $W_V$ and changing the query matrix to be $W_V^{-\top} W_K^\top W_Q$ does not change the behavior of the attention layer. Second, when the key and value matrices are equal, the key-value matrix is symmetric and, therefore, only requires $d(d+1)/2$ elements to be represented. This implies that, when the value matrix is invertible, the minimal number of hidden neurons our gated RNN needs to store key-values is in fact $d(d+1)/2 + d$. In Section 4, we show that learned RNNs find this solution.

Overall, the output gating requires $\mathcal{O}(d^2)$ input and output entries for the gated RNN to match a linear self-attention layer. The RNN thus requires $\mathcal{O}(d^4)$ parameters in total, with a lot of redundancy, significantly more than the $3d^2$ parameters of the linear self-attention layer. It comes as no surprise that numerous equivalent configurations exist within the gated RNN we study. For instance, linear gating is invariant under permutations of rows between its two matrices and under multiplication-division of these two rows by a constant. Left-multiplying $W_Q$ in the input gating by any invertible matrix $P$, and subsequently reading out the hidden neurons with $\lambda = 0$ through $\mathrm{repeat}(P^{-1}, d)$, also does not alter the network's output. Several other invariances exist, making exact weight retrieval nearly impossible.

## 3.3 IMPLICATIONS FOR OTHER CLASSES OF GATED RNNS

We conclude this section by commenting on whether similar insights hold for other gated RNNs architectures. The LRU architecture is close to (2) but only has output gating. Stacking two LRU layers on top of each other enables the output gating of the first layer to act as the input gating for the second layer and, therefore, implement the mechanism we highlighted in the previous sections to mimick attention. As noted in Section 2.2, LSTMs and GRUs are further away from our simplified gated RNN model. However, one single LSTM layer can implement linear self-attention, but stacked GRU layers cannot. Let us briefly summarize the argument behind these results. The LSTM layer has a sophisticated input gating mechanism that gates a candidate cell state based on the current input and previous state. The gate and the candidate cell state depend, among other things, on the current input. This mechanism can thus play a similar role to $g^{\mathrm{in}}$ and implement the key-value outer product. The recurrence of the cell state can be set to perfectly integrate key-values, by setting the forgetting gate accordingly. Finally, the output gate modulates the current cell state, which contains the accumulated key-values. Setting the output gate to encode the query enables computing the desired result. We note that the output gating differs from $g^{\mathrm{out}}$: it multiplies transformations of the cell state and the input instead of the input only. This property makes it possible to implement attention within one layers, where as two layers are required for our gated RNN model (2). While the GRU layer takes many of the computational elements from the LSTM, it cannot implement attention as it has no mechanism to compute multiply keys and values. We refer the reader to Appendix A for more details.

## 4 GATED RNNS LEARN TO MIMIC ATTENTION

We now demonstrate that gated RNNs learn to implement linear self-attention and comprehend how they do so. In this section, a student RNN is tasked to reproduce the output of a linear self-attention layer. Appendix B contains detailed descriptions of all experiments performed in this section. Importantly, each sequence is only presented once to the network.

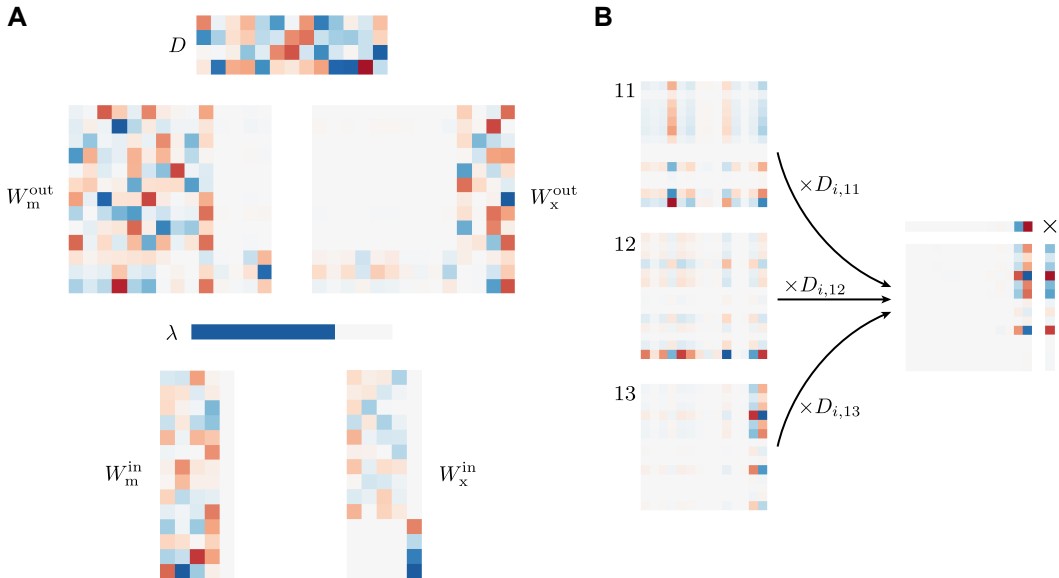

Figure 2: In our teacher-student experiment of Section 4.1 ($d = 4$), the structure of the weights of the RNN after learning matches the one of our compact construction, c.f. Section 3. **(A)** Only recurrent neurons with perfect memory ($\lambda = 1$, dark blue) or no memory at all ($\lambda = 0$, light grey) influence the output, consistently with the theory. The block structure almost perfectly match the one of our construction, c.f. Figure 1 **(B)** For each output coordinate $i$ of the network, the kernels generated by the last 3 lines (11, 12 and 13) of the output gating are linearly combined through the decoding matrix $D$ are all proportional to the same kernel, which can be generated in a way that is coherent with the structure of our construction. This way, the weights have effectively the exact block structure our construction has. In all the matrices displayed here, zero entries are shown in light grey, blue denotes positive entries, and red negative ones.

| Loss | Score KV | Score Q | Polynomial distance |
|------|----------|---------|---------------------|
| $4.97 \times 10^{-8}$ | $4.52 \times 10^{-8}$ | $2.06 \times 10^{-10}$ | $3.73 \times 10^{-4}$ |

Table 1: Gated RNNs implement the same function as a linear self-attention layer in our teacher-student experiment (Section 4.1). The KV and Q scores are equal to one minus the $R^2$ score of the linear regression that predicts key-values and queries from resp. the perfect memory neurons (those whose $\lambda = 1$) and perfect forget neurons ($\lambda = 0$). The polynomial distance is the L2 distance between the coefficients of the degree-4 polynomial that describes the instantaneous processing of the (optimal) linear self-attention layer and the trained RNN. Overall, this analysis reveals that the student RNN is functionally equal to the teacher attention layer both inside (training loss close to 0) and outside (the coefficients of the polynomials match) training distribution.

## 4.1 TEACHER IDENTIFICATION

In our first experiment, we train a student RNN ($|x| = 4$, $|h| = 100$ and $|y| = 4$) to emulate the behavior of a linear self-attention layer with weights sampled from a normal distribution and inputs $x_t$ sampled i.i.d. from a normal distribution. The low training loss, reported in Table 1, highlights that the student's in-distribution behavior aligns with the teacher's. However, this is insufficient to establish that the student implements the same function as the teacher. The strategy we adopt to show functional equivalence is as follows: First, we observe that only perfect memory neurons ($\lambda = 1$) and perfect forget neurons ($\lambda = 0$) influence the network output. Additionally, each of these groups of neurons receives all the information needed to linearly reconstruct resp. the key-values and the queries from the input. Finally, we show that the output gating and the decoder matrix accurately multiply accumulated key-values with current queries.

After the learning process, a significant proportion of the weights in the input and output gating and the readout becomes zeros. Consequently, we can eliminate neurons with input or output weights that are entirely zeros, thereby preserving the network's function. By doing so, we can remove 86 out of the 100 hidden neurons and 87 out of the 100 pre-readout neurons. After having permuted rows in the two gating mechanisms and reordered hidden neurons, we plot the resulting weights on Figure 2.A. Consistently with our construction, only recurrent neurons with $\lambda = 0$ or $\lambda = 1$ contribute to the network's output. The key-values neurons receive a polynomial of degree 2, as $g^{\mathrm{in}}$ is a bilinear form, without any term of degree 1 as the last column of $W_{\mathrm{m}}^{\mathrm{in}}$ and $W_{\mathrm{x}}^{\mathrm{in}}$ is equal to zero for those units. Similarly, the query neurons receive a polynomial of degree 1. The learning process discovers that it can only use $d(d+1)/2 = 10$ neurons to store key-values, similar to our optimal construction. We show in Table 1 that it is possible to linearly reconstruct the key-values from those 10 neurons perfectly, as well as the queries from the 4 query neurons. By combining this information with the fact that the $\lambda$s are zeros and ones, we deduce that the cumulative key-values $\sum_{t' \le t} v_{t'} k_{t'}^{\top}$ can be obtained linearly from the key-values' hidden neurons, and the instantaneous queries $q_t$ from the query neurons.

Additionally, the output gating combined with the linear readout can multiply the key-values with the queries. Since we have already confirmed that the temporal processing correctly accumulates key-values, our focus shifts to proving that the instantaneous processing of the gated RNN matches the one of the attention layer across the entire input domain. Given that both architectures solely employ linear combinations and multiplications, their instantaneous processing can be expressed as a polynomial of their input. The one of linear self-attention, $(W_V x)(W_K x)^{\top}(W_Q x)$, corresponds to a polynomial of degree 3, whereas the one of the gated RNN, $g^{\mathrm{out}}(g^{\mathrm{in}}(x))$, corresponds to one of degree 4. By comparing these two polynomials, we can compare their functions beyond the training domain. For every one of the four network outputs, we compute the coefficients of terms of degree 4 or lower of their respective polynomials and store this information into a vector. We then calculate the normalized Euclidean distance between these coefficient vectors of the linear self-attention layer and the gated RNN, and report the average over all 4 output units in Table 1. The evidence presented so far enables us to conclude that the student network has correctly identified the function of the teacher.

While the majority of the weights depicted in Figure 2.A conform to the block structure characteristic of our construction, the final three rows within the output gating matrices deviate from this trend. As shown in Figure 2.B, these three rows can be combined into a single row matching the desired structure. More details about this manipulation can be found in Appendix B.2.

## 4.2 IDENTIFICATION REQUIRES MILD OVERPARAMETRIZATION

The previous experiment shows that only a few neurons in a network of 100 hidden neurons are needed to replicate the behavior of a self-attention layer whose input size is $d$. We therefore wonder if identification remains possible when decreasing the number of hidden and pre-output gating neurons the student has. We observe that mild overparametrization, around twice as many neurons as the actual number of neurons required, is needed to reach identification. We report the results in Figure 3.A.

## 4.3 NONLINEARITY MAKES IDENTIFICATION HARDER

We now move away from our simplified class of gated RNNs and seek to understand how our findings apply to LSTMs, GRUs, and LRUs. We use the following architecture for those three layers: a linear embedding layer projects the input to a latent representation, we then repeat the recurrent layer once or twice, and finally apply a linear readout. While those layers are often combined with layer normalization, dropout, or skip connections in modern deep learning experiments, we do not include any of those here to stay as close as possible to the teacher's specifications. In an LRU layer, the input/output dimension differs from the number of different neurons; we here set all those dimensions to the same value for a fair comparison with LSTMs and GRUs. We compare these methods to the performance of our simplified gated RNNs, with both diagonal (as in Equation 2) and dense linear recurrent connectivity.

We report the results in Figure 4.A for inputs of dimension $d = 6$. While diagonal connectivity provides a useful inductive bias to learn how to mimic linear self-attention, it is not absolutely needed

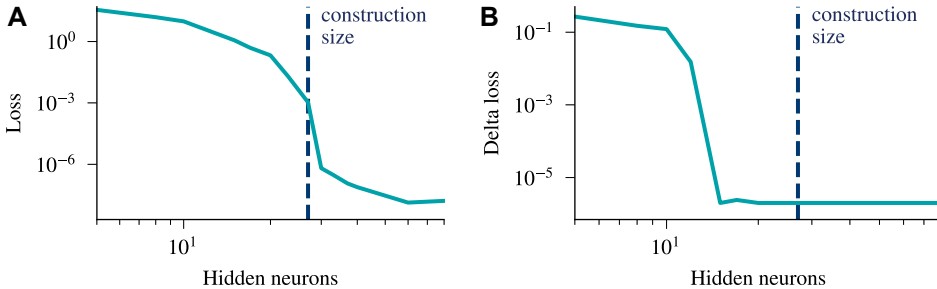

Figure 3: Gated RNNs learn compressed representation. **(A)** In the teacher-student experiment of Section 4 in which there is no clear structure within the attention mechanism that the RNN can extract, slight overparametrization is needed in order to identify the teacher. **(B)** In the linear regression task of Section 5, the linear attention mechanism needed to solve the task optimally has a sparse structure that the RNN leverages. Identification is thus possible with much smaller networks. The quantity reported in B is the difference between the prediction loss of the RNN and the loss obtained after one optimal step of gradient descent. We use the same input dimension $d = 6$ to make the two plots comparable.

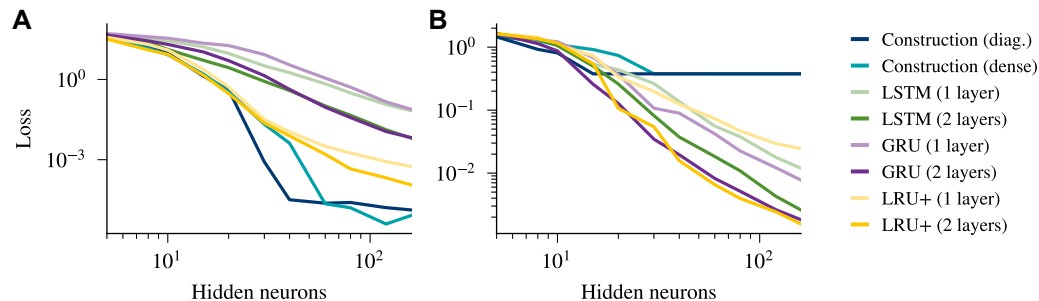

Figure 4: Comparison of the validation loss obtained by different gated recurrent networks architectures in **(A)** the teacher-student task of Section 4 and **(B)** the in-context linear regression task of Section 5. The construction baseline corresponds to the gated RNN of Eq. 2, with diagonal or dense connectivity. We use the default implementation of LSTMs and GRUs, and slightly modify the LRU architecture to reflect our construction better. Non-linearity improves the in-context learning performance but deteriorates the ability to mimic attention.

as changing the recurrence connectivity to be dense does not significantly affect performance. It is theoretically possible to identify the teacher with one LSTM layer. However, gradient descent does not find such a solution and the performance of LSTMs is close to that of GRUs that cannot implement attention. Motivated by the construction of Section 3, we slightly modify the LRU architecture (LRU+) and add a nonlinear input gating to the already existing output gating. We find that this modification significantly improves the ability of a LRU layer to mimic attention. Additionally, we confirm that multiplicative interactions are fundamental for mimicking attention: replacing gating with a 1-hidden layer MLP with the same number of parameters significantly deteriorates performance. Appendix B contains experiments that extensively compare different LRU architectures, as well as comparisons that take into account the number of parameters of the different architectures.

## 5 ATTENTION-BASED IN-CONTEXT LEARNING EMERGES IN TRAINED RNNS

The previous section shows that gated RNNs learn to replicate a given linear self-attention teacher. We now demonstrate that they can find the same solution as linear self-attention when both are learned. To that end, we study an in-context regression task in which the network is shown a few input-output pairs and later has to predict the output value corresponding to an unseen input. Linear self-attention is a particularly beneficial inductive bias for solving this task. When the input-output

| Term | RNN | GD |
|------|-----|-----|
| $x_1^2 y_1$ | $6.81 \times 10^{-2} \pm 8.52 \times 10^{-5}$ | $6.76 \times 10^{-2}$ |
| $x_2^2 y_1$ | $6.82 \times 10^{-2} \pm 6.40 \times 10^{-5}$ | $6.76 \times 10^{-2}$ |
| $x_3^2 y_1$ | $6.82 \times 10^{-2} \pm 5.56 \times 10^{-5}$ | $6.76 \times 10^{-2}$ |
| residual norm | $1.35 \times 10^{-3} \pm 1.97 \times 10^{-4}$ | $0$ |

Table 2: Gated RNNs implement gradient descent in the in-context linear regression task of Section 5 The coefficients of the instantaneous polynomial implemented by the first output unit of a trained RNN on the in-context linear regression task match one optimal step of gradient descent, averaged over 4 seeds. The residual norm measures the norm of the polynomial coefficients, excluding the ones appearing in the table. Those coefficients are all vanishingly small. The optimal GD learning rate is obtained analytically ($\eta^* = (T + d_x - 1/5)^{-1}$), c.f. Appendix C.2.

mapping is linear, von Oswald et al. (2023) have shown that linear self-attention implement one step of gradient descent.

## 5.1 IN-CONTEXT LINEAR REGRESSION

Linear regression consists in estimating the parameters $W^* \in R^{d_y \times d_x}$ of a linear model $y = W^* x$ from a set of observations $\{(x_t, y_t)\}_{t=1}^T$ that satisfy $y_t = W^* x_t$. The objective consists in finding a parameter $\hat{W}$ which minimizes the squared error loss $L(W) = \frac{1}{2T} \sum_{t=1}^T \|y_t - W x_t\|^2$. Given an initial estimate of the parameter $W_0$, one step of gradient descent on $L$ with learning rate $T\eta$ yields the weight change

$$\Delta W_0 = \eta \sum_{t=1}^T (y_t - W_0 x_t) x_t^\top. \tag{3}$$

In the in-context version of the task, the observations $(x_t, y_t)_{1 \le t \le T}$ are provided one after the other to the network, and later, at time $T+1$, the network is queried with $(x_{T+1}, 0)$ and its output regressed against $y_{T+1}$. Under this setting, von Oswald et al. (2023) showed that if all bias terms are zero, a linear self-attention layer learns to implement one step of gradient descent starting from $W_0 = 0$ and predict through

$$\hat{y}_{T+1} = (W_0 + \Delta W_0) x_{T+1} = \eta \sum_{t=1}^T y_t x_t^\top x_{T+1}. \tag{4}$$

In the following, we show that gated RNNs also learn to implement the same algorithm and leverage the sparse structure of the different attention matrices corresponding to gradient descent to learn a more compressed representation than the construction one.

## 5.2 GATED RNNs LEARN TO IMPLEMENT GRADIENT DESCENT

We now train gated RNNs as in Equation 2 to solve the in-context linear regression task, see Appendix C.1 for more details. We set the number of observations to $T = 12$ and set the input and output dimensions to 3 so that $d = 6$. Once learned, the RNN implements one step of gradient descent with optimal learning rate, which is also the optimal solution one layer of linear self-attention can find (Mahankali et al., 2023). Several pieces of evidence back up this claim: the training loss of RNN after training (0.0945) is almost equal to the one of an optimal step of gradient descent (0.0947) and the trained RNN implements the same instantaneous function, as the polynomial analysis of Table 2 reveals.

Linear self-attention weights implementing gradient descent have a very specific sparse structure (von Oswald et al., 2023). In particular, many key-values entries are always 0, so the construction contains many dead neurons. This leads us to wonder whether gated RNNs would pick up this additional structure and learn compressed representations. To test that, we vary the gated RNN size and report in Figure 3.B the difference between the final training loss and the loss obtained after one optimal gradient descent step. We observe a similar phase transition than in the teacher-student experiment, this time happening for a much smaller number of neurons than our construction specifies. Gated RNNs thus learn a more compressed representation than the one naively mimicking

self-attention. This result provides some hope regarding the poor $\mathcal{O}(d^4)$ scaling underlying our construction: in situations that require an attention mechanism with sparse $(W_V, W_K, W_Q)$ matrices, gated RNNs can implement attention with far fewer neurons. A precise understanding of how much compression is possible in practical scenarios requires further investigation.

### 5.3 NONLINEAR GATED RNNS ARE BETTER IN-CONTEXT LEARNERS THAN ONE STEP GRADIENT DESCENT

Finally, as a side question, we compare the ability to learn in context of the nonlinear gated RNN architectures that are LSTMs, GRUs and LRUs. Although not the main focus of our paper, this allows us to put our previous results in perspective. In particular, we are interested in understanding if similarity with attention correlates with in-context learning performance, as attention has been hypothesized to be a key mechanism for in-context learning (Olsson et al., 2022; Garg et al., 2022; von Oswald et al., 2023). We report our comparison results in Figure 4.B, measuring the loss on weights $W^*$ drawn from a distribution with double the variance of the one used to train the model.

Overall, we find that nonlinearity greatly helps and enables nonlinear gated RNN architectures to outperform one gradient descent step when given enough parameters, suggesting that they implement a more sophisticated mechanism. Surprisingly, while the GRU is the architecture that is the furthest away from attention, it performs the best in the task. Better understanding the mechanisms underlying this ability requires future work. Within the different LRU layers we compare, we find a high correlation between in-context learning abilities and closeness to attention, c.f. Figure 5 in the Appendix. In particular, we observe a massive performance improvement from the vanilla LRU architecture to the ones additionally including input gating to match our construction more closely.

## 6 DISCUSSION

Our study reveals a closer conceptual relationship between RNNs and Transformers than commonly assumed. We demonstrate that gated RNNs can theoretically and practically implement linear self-attention, bridging the gap between these two architectures. Moreover, while Transformers have been shown to be powerful in-context learners (Brown et al., 2020; Chan et al., 2022), we find that RNNs excel in toy in-context learning tasks and that this performance is partly uncorrelated with the architecture inductive bias toward attention. This highlights the need for further investigations on the differences between RNNs and Transformers in controlled settings, as also advocated by Garg et al. (2022).

Our results partly serve as a negative result: implementation of attention is possible but requires squaring the number of parameters attention has. We have shown that gated RNNs can leverage possible compression, but understanding whether real-world attention mechanisms lie in this regime remains an open question. Yet, our work is of current practical relevance as it provides a framework that can guide future algorithmic developments, as we exemplify in Appendix A.4. Bridging the gap between Transformers' computational power and RNNs' inference efficiency is a thriving research area (Fournier et al., 2023), and the link we made facilitates interpolation between those two model classes.

Finally, our work carries implications beyond deep learning. Inspired by evidence from neuroscience supporting the existence of synaptic plasticity at different timescales, previous work (Schmidhuber, 1992; Ba et al., 2016; Miconi et al., 2018) added a fast Hebbian learning rule, akin to linear self-attention, to slow synaptic plasticity with RNNs. We show that, to some extent, this mechanism already exists within the neural dynamics, provided that the response of neurons can be multiplicatively amplified or shut-off in an input-dependent manner. Interestingly, several single-neuron and circuit level mechanisms have been experimentally identified which could support this operation in biological neural networks (Silver, 2010). We speculate that such multiplicative mechanisms could be involved in implementing self-attention-like computations in biological circuitry.

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

## A  GATED RNNS AND LINEAR SELF-ATTENTION

In this section, we compare our simplified gated RNN model, linear self-attention, and nonlinear gated RNN models (LSTMs, GRUs and LRUs). We recall that the key ingredients of our simplified gated RNNs defined as

$$h_{t+1} = \lambda \odot h_t + g^{\text{in}}(x_t), \quad y_t = Dg^{\text{out}}(h_t), \tag{5}$$

are the diagonal linear recurrence and the input and output gating. The input gating serves as a way to generate the key-values of linear self-attention, which will then be accumulated in the hidden recurrent units and combined with queries within the output gating.

Table 3 summarizes how many layers of LRUs, LSTMs and GRUs are needed to exactly implement our simplified class of gated RNNs and linear self-attention. We provide more details below.

|  | Simplified gated RNN | Linear self-attention |
|---|---|---|
| LRU | 2 | 2 |
| LRU In-Out | 1 | 1 |
| LRU In-Out (MLP) | – | – |
| LSTM | 2 | 1 |
| GRU | – | – |

Table 3: Number of layers needed for different RNN layers to exactly implement our simplified class and linear self-attention.

### A.1  LRU

An LRU layer (Orvieto et al., 2023b) consists of a recurrent state $h_t$ and some instantaneous post-processing. Its recurrent state is updated as

$$h_{t+1} = \lambda \odot h_t + \gamma \odot (Bx_{t+1}) \tag{6}$$

and its output $y_t$ is computed with

$$\tilde{y}_{t+1} = \text{Re}[Ch_t] + Dx_{t+1} \tag{7}$$
$$y_{t+1} = \sigma(W_{\text{m}}\tilde{y}_{t+1}) \odot (W_{\text{x}}\tilde{y}_{t+1}). \tag{8}$$

In the equations above, $h_{t+1}$, $B$ and $C$ are complex-valued, Re denotes the real part of a complex number, and $\sigma$ is the sigmoid function. The transformation nonlinear transformation between $y_{t+1}$ and $\tilde{y}_{t+1}$ is called a gated linear unit (GLU) and was introduced in Dauphin et al. (2017). Additionally, $\lambda$ and $\gamma$ are parametrized exponentially:

$$\lambda = \exp(-\exp(\nu^{\log}) + i\exp(\theta^{\log})) \text{ and } \gamma = \exp(\gamma^{\log}). \tag{9}$$

The LRU layer detailed above comprises two central computational mechanisms: a linear recurrence coupled with a GLU serving as nonlinear output gating. The recurrence is here complex-valued, but we only need the real part of it for our purposes. Assuming that the sigmoid can be linearized, our class of gated RNNs can be implemented using two layers by letting the output gating of the first layer serve as input gating. We are now left with linearizing the sigmoid. To achieve this, we double the number of output neurons of the GLU and require small weights in $W_{\text{m}}$, that can for example, be compensated by large weights in $W_{\text{m}}$. Under this regime, we have $\sigma(W_{\text{m}}x) \odot (W_{\text{x}}x) \approx (1/2 + W_{\text{m}}x) \odot (W_{\text{x}}x)$. Half of the neurons require identical weights as the target linear gating (up to a proportional factor), half should have $W_{\text{m}} = 0$ and the same $W_{\text{x}}$ as target linear gating. The $1/2W_{\text{x}}x$ term that comes from the second half of the neurons can be subtracted from the first half of the neurons in a subsequent linear transformation, thereby yielding the desired result.

In our experiments, we consider two additional variations of the LRU layer that can implement our class of gated RNNs and/or linear self-attention using only one layer. The LRU In+Out variation has an additional nonlinear input gating mechanism compared to the original version (LRU Out) that modifies the input before the recurrent part of the layer. The LRU In+Out (MLP) replaces the GLU in the LRU In-Out variation by a 1-hidden layer MLP, keeping the number of parameters fixed. The LRU In-Out variation can implement both linear self-attention and our class of gated RNNs in one layer, whereas LRU In-Out (MLP) cannot, as it does not have any multiplicative interactions.

## A.2 LSTM

An LSTM cell (Hochreiter & Schmidhuber, 1997) has two recurrent states: the hidden state $h_t$ and the cell state $c_t$. They are updated as follows.

$$f_{t+1} = \sigma(U_f x_{t+1} + V_f h_t + b_f) \tag{10}$$
$$\tilde{c}_{t+1} = \tanh(U_c x_{t+1} + V_c h_t + b_c) \tag{11}$$
$$g_{t+1} = \sigma(U_g x_{t+1} + V_g h_t + b_g) \tag{12}$$
$$c_{t+1} = f_{t+1} \odot c_t + g_{t+1} \odot \tilde{c}_{t+1} \tag{13}$$
$$o_{t+1} = \sigma(U_o x_{t+1} + V_o h_t + b_o) \tag{14}$$
$$h_{t+1} = o_{t+1} \odot \tanh(c_{t+1}). \tag{15}$$

Here, $f_t$ is the cell state forget gate, $\tilde{c}_t$ the cell state update candidate, $g_t$ the cell state update candidate gate, $o_t$ the output gate, and $\sigma$ the sigmoid function applied element-wise.

First, we show that one single LSTM layer can implement linear self-attention, by using $g_{t+1} \odot \tilde{c}_{t+1}$ as a way to compute key-values and $c$ to aggregate them, $f_{t+1}$ and use $o_{t+1}$ for the query. We provide the corresponding weights in the table below, ignoring all the nonlinearities except $\sigma$ in the $f$ computation. Note that, compared to our simplified gated RNN class, we do not need to include neurons that forget their last state ($\lambda = 0$) here as the output gate directly provides the query to the output. Finally, linearizing the $\tanh$ function requires small $U_c$ weights that can later be compensated by large decoder weights, and ways to linearize the sigmoid were discussed in the previous section.

Implementing a gated RNN as in Equation 2 can be done by using two layers: in the first layer $g_{t+1} \odot \tilde{c}_{t+1}$ serves as input gating, $f_{t+1}$ corresponds to $\lambda$, and, in the second layer, $g_{t+1} \odot \tilde{c}_{t+1}$ serves as output gating. Table 4 provides one set of such weights. This ignores the linearization trick for the $\tanh$ in $\tilde{c}$ and the sigmoid in $g_{t+1}$.

| | Layer 1 | | |
|---|---|---|---|
| | $U$ | $V$ | $b$ |
| $f$ | $0$ | $0$ | $+\infty$ |
| $\tilde{c}$ | $\tilde{W}_K$ | $0$ | $0$ |
| $g$ | $\tilde{W}_V$ | $0$ | $0$ |
| $o$ | $\tilde{W}_Q$ | $0$ | $0$ |

| | Layer 1 | | | Layer 2 | | |
|---|---|---|---|---|---|---|
| | $U$ | $V$ | $b$ | $U$ | $V$ | $b$ |
| $f$ | $0$ | $0$ | $\sigma^{-1}(\lambda)$ | $0$ | $0$ | $-\infty$ |
| $c$ | $W_m^{in}$ | $0$ | $0$ | $W_m^{out}$ | $0$ | $0$ |
| $g$ | $W_x^{in}$ | $0$ | $0$ | $W_x^{out}$ | $0$ | $0$ |
| $o$ | $0$ | $0$ | $+\infty$ | $0$ | $0$ | $+\infty$ |

Table 4: LSTM weight configuration that matches a linear self-attention layer (left) and a gated RNN as in Equation 2 (right). This presumes that the activation functions in $\tilde{c}$, $g$ and $o$ are linear. We use $\tilde{W}$ to denote the value, key and query matrices transformed in a similar way to what we did in Figure 1.

## A.3 GRU

A GRU cell (Cho et al., 2014) has a hidden state $h_t$, updated through

$$r_{t+1} = \sigma(U_r x_{t+1} + V_r h_t + b_r) \tag{16}$$
$$\tilde{h}_{t+1} = \tanh(U_h x_{t+1} + V_h(r_{t+1} \odot h_t) + b_h) \tag{17}$$
$$z_{t+1} = \sigma(U_z x_{t+1} + V_z h_t + b_z) \tag{18}$$
$$h_{t+1} = (1 - z_{t+1}) \odot h_t + z_{t+1} \odot \tilde{h}_{t+1} \tag{19}$$

where $r_t$ is the reset gate, $z_t$ is the update gate, $\tilde{h}_t$ the update candidate, and $\sigma$ is the sigmoid function.

Here, stacking multiple GRUs on top of each other does not enable the implementation of any network from our class of gated RNNs nor linear self-attention layers. One layer can implement diagonal linear recurrence by linearizing the $\tanh$, having $z_{t+1} = 1$ and $r_{t+1} = \lambda$. However,

implementing a gating mechanism of the form $g(x) = (W_\mathrm{m}x \odot W_\mathrm{x}x)$ is not possible[1]: we would need to use $z_{t+1}$ to implement one branch of the gating and $\tilde{h}_{t+1}$ the other but, given that $z_{t+1} \neq 0$, the previous hidden state $h_t$ influence the result.

## A.4 CAN LINEAR SELF-ATTENTION IMPLEMENT GATED RECURRENT NETWORKS?

Throughout the paper, we mainly focus on understanding whether diagonal gated RNNs implement linear self-attention. In this section, we ask the opposite question: can linear self-attention layers can implement gated recurrent networks. The answer is that attention layers as we defined in Section 2.1 cannot, because it can only perfectly integrate inputs or send the current one (thus $\lambda = 0$ or $\lambda = 1$). However, adding a mechanism akin to weight decay bridges the gap. In particular, we will describe how the output $y_t$ of a such a linear self-attention layer can satisfy a recurrence relationship of the form $y_{t+1} = \lambda \odot y_t + x_t$. To do so, we consider the following attention layer:

$$y_t = \left( \sum_{t'=1}^{t} \Gamma_{t-t'} \odot (W_V x_{t'} + b_V)(W_K x_{t'} + b_K)^\top \right)(W_Q x_t + b_Q) \tag{20}$$

where $\Gamma_{t-t'}$ is a matrix of size $d \times d$ in which all entries of the $i$-th row have value $(1 - \gamma_i)^{t-t'}$. The $\gamma$ term can be interpreted as a weight decay: if we note

$$W_t^\mathrm{ff} := \left( \sum_{t'=1}^{t} \Gamma_{t'-t} \odot (W_V x_{t'})(W_K x_{t'})^\top \right), \tag{21}$$

we have

$$W_{t+1}^\mathrm{ff} = W_t^\mathrm{ff} + (W_V x_{t+1} + b_V)(W_K x_{t+1} + b_K)^\top - \Gamma_1 W_t^\mathrm{ff}. \tag{22}$$

Now, we set the value, key and query matrices and biases to $W_V = \mathrm{Id}, b_V = 0, W_K = 0, b_K = 1, W_Q = 0, b_Q = 1/d$ and $1 - \gamma = \lambda$. This way, we have

$$y_{t+1} = \frac{1}{d} W_{t+1}^\mathrm{ff} 1 \tag{23}$$

$$= \frac{1}{d} \left( \Gamma_1 \odot W_t^\mathrm{ff} + x_{t+1} 1^\top \right) 1 \tag{24}$$

$$= \left( \Gamma_1 \odot W_t^\mathrm{ff} \right) 1 + x_{t+1} \tag{25}$$

$$= \lambda \odot y_t + x_{t+1} \tag{26}$$

In the last line, we use the structure of $\Gamma_1$ and the value of $\gamma$. Biases terms are crucial to make this link: without them $W_t^\mathrm{ff}$ would be a polynomial with only degree 2 coefficients and the equivalence would not be possible. The gating mechanism within networks described in Equation 2 can also be implemented by forgetting $(1 - \gamma = 0)$ and having the key-value taking care of the multiplication.

This analysis reveals the importance of weight decay to implement recurrent neural network like computations with a wide range of timescales. Adding complex-valued weight decay to linear self-attention layers makes them closer to state-of-the-art recurrent neural networks architecture (Orvieto et al., 2023b; Smith et al., 2023) for capturing long-range dependencies. Therefore, such a modification might boost the performance of attention layers on benchmarks testing these properties, such as the Long Range Arena (Tay et al., 2020). Interestingly, this view can partly explain the great empirical performance of the RWKV (Peng et al., 2023), which features a similar mechanism to weight decay. Overall, the analysis we conducted in this section examplify how the connection between RNNs and attention layers we made in this paper can be used to guide development of future architectures.

## B TEACHER-STUDENT

### B.1 EXPERIMENTAL DETAILS

For all experiments in Section 4, we train the student for almost one million training iterations on sequences of length 32 and a batch size of 64 (50000 training examples per epoch, 1000 epochs).

---

[1]When the $\mathrm{tanh}$ is replaced by Id, it is possible to achieve so by having $h_t \ll \tilde{h}_{t+1}$ and correcting for the exponential growth in the next layer.

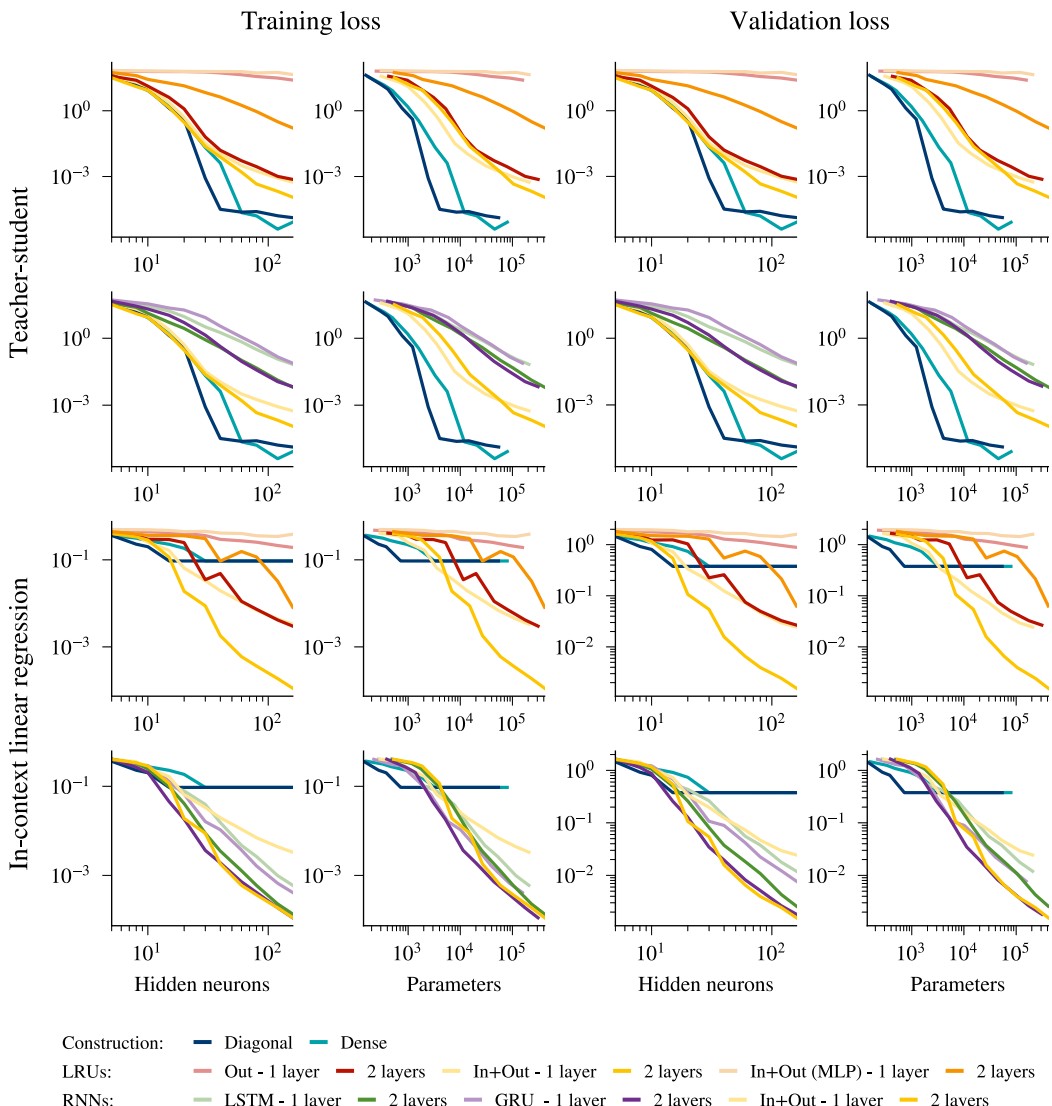

Figure 5: Extensive comparison between the different architectures. Compared to Figure 4, we consider different versions of the LRU here, plot the loss as the function of the number of parameters, and include both training and validation losses. Those two losses are almost (up to some sampling noise) for the teacher-student task but are different for the in-context linear regression task because we change the $W^*$ distribution in the validation set.

We use the AdamW (Loshchilov & Hutter, 2019) optimizer with a cosine annealing learning rate scheduler. The initial learning rate is set at $10^{-3}$, scheduled to anneal down to $10^{-6}$ by the end of training and a weight decay of $10^{-4}$ is applied to all parameters except the recurrent ones $\lambda$ in the experiment of Section 4.1. To ensure that the hidden states do not explode, we ensure that $\lambda$ stays within $[0, 1]$ by employing the exponential parametrization described in Appendix A.1 (we only keep the $\nu$ part as $\lambda$ takes real values here).

In Figure 5, we add more results to the architecture comparison we did in Figure 4. In particular, we compare the three different types of LRU we mentioned in Appendix A.1, and observe that adding an input GLU improves LRUs ability to mimic linear self-attention within one layer, but also with several layers.

## B.2 COMPRESSION OF THE LEARNED OUTPUT GATING WEIGHTS

In Figure 2, we show that the gating weight matrices have a structure that is close to the one of our construction, except for three different rows (11, 12, and 13). We claim they can be reduced to a single row; we now provide details justifying it.

Therefore, our objective is to demonstrate that these three rows are functionally equivalent to a single row with the expected structure and to gain insights into the invariances inherent to the gating mechanism we study in this paper along the way. The initial step toward achieving this entails examining the influence of these three rows on the $i$-th coordinate of the network's output:

$$\sum_{j=11}^{13} D_{i,j} g^{\text{out}}(h)_j = \sum_{j=11}^{13} D_{i,j}(W_{\text{m},j}^{\text{out}}x)(W_{\text{x},j}^{\text{out}}x) = x^\top \left( \sum_{j=11}^{13} D_{i,j} W_{\text{m},j}^{\text{out}} W_{\text{x},j}^{\text{out}\top} \right) x. \quad (27)$$

This contribution can be interpreted as a quadratic form whose kernel is a weighted sum of rank-1 kernels defined by the rows of the output gating matrices. In Figure 2.C, we plot the obtained kernel for one of the output components. Crucially, the resulting kernel for the four output units are all proportional to one another and is of rank-1. We can thus reduce the three neurons (11, 12 and 13) to one. Furthermore, the two vectors whose outer product yields the resulting kernel now mirror the construction's structure. One of these two vectors exclusively accesses query neurons while the other reads key-value neurons, as seen in Figure 2.C. As usually occurs with this kind of manipulation (Martinelli et al., 2023), merging the neurons slightly increases the loss, but original loss levels can be recovered after fine-tuning.

## C IN-CONTEXT LINEAR REGRESSION

### C.1 EXPERIMENTAL DETAILS

In the in-context linear regression experiment, each sequence is a task characterized by a unique $W^*$. The weight matrix $W^*$ entries are sampled i.i.d. from a normal distribution $\mathcal{N}(0, \frac{1}{3})$. Each element of the sequence is of the form $(x_t, W^* x_t)$. The entries of the inputs $(x_t)_{t=1}^{T+1}$ are sampled i.i.d. from the uniform distribution $\mathcal{U}(-\sqrt{3}, \sqrt{3})$. During the validation phase, we draw tasks from a different distribution, $W_{ij}^* \sim \mathcal{N}(0, \frac{2}{3})$ to highlight the generalization abilities of the learned models. We train the model with the same optimization scheme described in Appendix B.1, except that we use a smaller number of training iterations, totaling $300,000$. By default, we use gated RNNs with 80 hidden neurons.

### C.2 OPTIMAL LEARNING RATE FOR ONE-STEP GRADIENT DESCENT

Let $X \in \mathbb{R}^{d_x \times n}, W \in \mathbb{R}^{d_y \times d_x}$ random variables such that all entries of $X$ are sampled i.i.d. from a centered uniform distribution with variance $\sigma_x^2$, and those of $W$ i.i.d. from some centered distribution with finite variance $\sigma_W^2$. We set $Y = WX$. Let $x \in \mathbb{R}^{d_y}$ a column vector, whose entries are sampled from the same distribution as those of $X$, and $y = Wx$.

The goal of this section is to analytically derive the optimal learning rate for the in-context linear regression task, that is to find $\eta$ which minimizes

$$\mathcal{L}(\eta) = \frac{1}{2} \mathbb{E}_{X,W,Y,x,y} \left[ \|y - \hat{W}(\eta, X, Y)x\|^2 \right] \quad (28)$$

where $\hat{W}(X, Y)$ is the result of one gradient descent step starting from 0 with learning rate $\eta$ on the loss $W \mapsto \frac{1}{2}\|Y - WX\|^2$. The calculation is presented in a more general form in Mahankali et al. (2023). We include it here as we additionally provide a simple formula for exact optimal learning rate value.

Plugging in the analytical expressions for $y$ and $\hat{W}$, we get

$$\mathcal{L}(\eta) = \frac{1}{2}\mathbb{E}_{X,W,Y,x,y}\left[\|y - \eta Y X^\top x\|^2\right] \tag{29}$$

$$= \frac{1}{2}\mathbb{E}_{X,W,x}\left[\|Wx - \eta W X X^\top x\|^2\right] \tag{30}$$

$$= \frac{1}{2}\mathbb{E}_{X,W,x}\left[\|W(I - \eta X X^\top)x\|^2\right] \tag{31}$$

We want to minimize $\mathcal{L}$, i.e. look for $\eta^*$ that satisfies $\partial_\eta \mathcal{L}(\eta^*) = 0$. We have

$$\partial_\eta \mathcal{L}(\eta) = \mathbb{E}_{X,W,x}\left[\left(W(I - \eta X X^\top)x\right)^\top W X X^\top x\right] \tag{32}$$

$$= \operatorname{Tr}\mathbb{E}_{X,W,x}\left[(I - \eta X X^\top)W^\top W X X^\top x x^\top\right] \tag{33}$$

$$= \sigma_x^2 \operatorname{Tr}\mathbb{E}_{X,W}\left[(I - \eta X X^\top)W^\top W X X^\top\right] \tag{34}$$

$$= \sigma_x^2 \operatorname{Tr}\mathbb{E}_{X,W}\left[X X^\top(I - \eta X X^\top)W^\top W\right] \tag{35}$$

$$= \sigma_x^2 \sigma_W^2 \operatorname{Tr}\mathbb{E}_X\left[X X^\top(I - \eta X X^\top)\right] \tag{36}$$

In the first equation, we use that $\mathbb{E}[a^\top b] = \operatorname{Tr}\mathbb{E}[ba^\top]$. Third and fifth ones make use of $\mathbb{E}_x[xx^\top] = \sigma_x^2\operatorname{Id}$ and $\mathbb{E}_W[WW^\top] = \sigma_W^2\operatorname{Id}$. Having $\partial_\eta \mathcal{L}(\eta^*) = 0$ is then equivalent to

$$\eta^\star := \frac{\operatorname{Tr}\mathbb{E}_X[X X^\top]}{\operatorname{Tr}\mathbb{E}_X[X X^\top X X^\top]}. \tag{37}$$

This result shows that only the distribution of the learning data matters. Let us compute this quantity. We have $\mathbb{E}_X[X X^\top] = n\sigma_X^2\operatorname{Id}$ so we are left with computing $\mathbb{E}_x[X X^\top X X^\top]$. Using that entries of $X$ are i.i.d., we get

$$\operatorname{Tr}\mathbb{E}_X[X X^\top X X^\top] = d_x\mathbb{E}_X\left[\sum_i\left(\sum_t x_{i,t}x_{1,t}\right)^2\right] \tag{38}$$

$$= d_x\mathbb{E}_X\left[\left(\sum_t x_{1,t}^2\right)^2\right] + d_x(d_x - 1)\mathbb{E}_X\left[\left(\sum_t x_{1,t}x_{2,t}\right)^2\right] \tag{39}$$

$$= d_x\mathbb{E}_X\left[\sum_t x_{1,t}^4 + \sum_{t\neq t'} x_{1,t}^2 x_{1,t'}^2\right] + d_x(d_x - 1)\mathbb{E}_X\left[\sum_t x_{2,t}^2 x_{1,t}^2\right] \tag{40}$$

$$= \frac{9}{5}n d_x\sigma_x^4 + n(n - 1)d_x\sigma_x^4 + n(d_x - 1)\sigma_x^4 \tag{41}$$

$$= n d_x\sigma_x^4\left(n + d_x - \frac{1}{5}\right) \tag{42}$$

because the fourth moment of a centered uniform distribution is $\frac{9}{5}\sigma_x^4$. Putting everything together, we finally have

$$\eta^* = \frac{1}{\sigma_x^2(n + d_x - \frac{1}{5})}. \tag{43}$$

## D  SOFTWARE

We run our experiments using the Jax (Bradbury et al., 2018) Python framework, using the Flax (Heek et al., 2023) library for neural networks. We base our code base on the Minimal-LRU (Zucchet et al., 2023a) repository. Data analysis and visualization were done using Numpy (Harris et al., 2020), Scikit-learn (Pedregosa et al., 2011) and Matplotlib (Hunter, 2007).

