# OpenReview forum: "Gated recurrent neural networks discover attention"
_ICLR.cc/2024/Conference — Submitted to ICLR 2024_

### Official Review · Reviewer_1u2u · 2023-10-29

**Soundness:** 3 good
**Presentation:** 3 good
**Contribution:** 3 good
**Rating:** 6
**Confidence:** 3

**Summary:**

This work analyzes recent developments in linear gated RNN/SSMs in the context of linear attention. The work shows how to construct a set of parameters in gated RNNs that can exactly implement linear self-attention. The paper also shows how LSTMs can be constructed in this way as well, but GRUs cannot. Synthetic experiments are performed that show the gated RNNs can learn the attention construction in a student-teacher setup. Experiments are then performed that show gated RNNs can find the linear attention solution when trained on an in-context learning linear regression task.

**Strengths:**

- Overall the paper provides an interesting analysis of the connection between gated linear RNNs and linear self-attention.

- The paper makes a nice connection that shows how linear self-attention can be exactly implemented within the weights of a gated RNN (if a quadratic increase in parameters is used). The investigation into LSTMs and GRUs is also interesting.

- The experiments flow nicely from showing it is possible for the gated RNNs to learn the linear attention solution in a teacher-student setup, to then showing that when trained from scratch they can also learn the solution in the linear regression task. The additional experiments related to overparameterization and nonlinearities and identification are also interesting.

**Weaknesses:**

- Figure 1 is helpful, but the paper would benefit from also formalizing the construction in equations, either in the main paper in Section 3.1 or in the Appendix. I found myself having to stare at Figure 1 and the description in Section 3.1 longer than probably necessary, whereas I think a bit of math (in particular with dimensions clearly defined) along with the figure and description would make this much easier to see.

- The experiments are demonstrative, but very toy, and have a lack of diversity. This is mostly ok for this type of paper, but it is unclear how well the results generalize. Perhaps analyzing and experimenting with additional tasks could be helpful. An additional toy task that might have been interesting is the associative recall/inductive head tasks from https://arxiv.org/pdf/2302.10866.pdf, https://arxiv.org/pdf/2212.14052.pdf, https://arxiv.org/abs/2209.11895. In particular, the H3 work also proposes a construction of how softmax attention can solve these tasks. Given that these tasks are of great interest to those studying language modeling with linear RNNs/SSMs, connecting with this prior work might broaden the audience of this work.

- More discussion and analysis around some of the results would strengthen the paper.
   - In particular, the compression result from Figure 3.B where the gated RNNs can solve the linear regression task with a size smaller than the theoretical construction size. Are there other tasks where this is not the case? E.g. perhaps the associative recall task from the point above? More analysis and experimentation around this point would strengthen the paper
  - While potentially more difficult, I would have also appreciated more discussion, analysis, experiments around the GRU results presented in Figure 4.B, since it does so well despite not reflecting the linear attention solution. Again, perhaps an additional experiment might be insightful.

**Questions:**

See weaknesses.

---

> ### Author Response · Authors · 2023-11-17
>
> We thank the reviewer for their on-point feedback.
>
> -   Here is the precise description of the values the different weight matrices take. The key-values are stored in the first $d^2$ recurrent neurons and the queries in the last $d$ ones (indices $d^2+1$ to $d^2 + d$.
>
>     - For the input gating: $W\_\mathrm{x}^\mathrm{in}$ and $W^\mathrm{in}\_\mathrm{g}$ are matrices of size $(d^2 + d) \times (d+1)$ with
>
>     $$ (W\_\mathrm{x}^\mathrm{in})\_{i,j} = \left \\{ \begin{array}{ll} {(W_V)}\_{i/d,j} &\text{ if } j \leq d \text{ and } i \leq d^2 \\\\ {(W\_Q)}\_{i-d^2,j} & \text{ if } j \leq d \text{ and } i > d^2 \\\\ 0 & \text{ otherwise} \end{array} \right . $$
>
>     $$ {(W\_\mathrm{g}^\mathrm{in})}\_{i,j} = \left \\{ \begin{array}{ll} {(W_K)}\_{i\,\mathrm{mod}\,d,j} &\text{ if } j \leq d \text{ and } i \leq d^2 \\\\ 1 & \text{ if } j = d+1 \text{ and } i > d^2\\\\ 0 & \text{ otherwise} \end{array} \right . $$
>
>     - For the recurrent neurons: $\lambda$ is a vector of size $d^2 + d$ with
>
>     $$ \lambda\_i = \left \\{ \begin{array}{ll} 1 & \text{ if } i \leq d^2\\\\ 0 & \text{ otherwise } \end{array} \right . $$
>
>     - For the output gating: $W^\mathrm{out}\_\mathrm{x}$ and $W^\mathrm{out}\_\mathrm{g}$ are matrices of size $d^2 \times (d^2 + d)$ with
>
>     $$ {(W\_\mathrm{x}^\mathrm{out})}\_{i,j} = \left \\{ \begin{array}{ll} 1 &\text{ if } j \leq d \text{ and } i = j \\\\
>      0 & \text{ otherwise} \end{array} \right . $$
>
>     $$ {(W\_\mathrm{g}^\mathrm{out})}\_{i,j} = \left \\{ \begin{array}{ll} 1 & \text{ if } j > d^2 \text{ and } i = j \\, \mathrm{mod} \\, d \\ 0 & \text{ otherwise} \end{array} \right . $$
>
>     For the readout matrix: $D$ is of size $d \times d^2$ with
>
>     $$ D\_{i,j} = \left \\{ \begin{array}{ll} 1 & \text{ if } i = j / d\\\\ 0 & \text{ otherwise } \end{array} \right . $$
>
>     We will include this in the appendix of the next version of the paper.
>
> -   The associative recall task is a great suggestion. We included promising preliminary results on this task in the global response.
>
> -   For the first point, we refer the reviewer to our global response. Regarding the second point, we agree with the reviewer that this is a very intriguing result. We have been investigating the reasons behind this performance but so far remained unsuccessful. We believe that this is an interesting direction for future work.
>
>
> We believe the new experimental results we provide following the reviewer advice greatly improve the paper and we hope that they may convince the reviewer to reconsider their score.

---

> > ### Comment · Reviewer_1u2u · 2023-11-21
> >
> > I thank the authors for the clarification and the additional associative recall experiment and the new analysis and discussions. I think this will be an interesting paper for the community to read and I have increased my score.

---

> > > ### Author Response · Authors · 2023-11-21
> > >
> > > We thank the reviewer once more for the great suggestions and for reevaluating their score.

---

### Official Review · Reviewer_pjc5 · 2023-10-31

**Soundness:** 4 excellent
**Presentation:** 4 excellent
**Contribution:** 2 fair
**Rating:** 5
**Confidence:** 5

**Summary:**

This paper provides a construction proof to demonstrate that gated linear recurrent units can learn linear autoregressive self-attention exactly.  The first experimental results (section 4) show that the theoretical result holds in practice: a GLRU network trained as the student to a linear self-attention network learns to imitate its teacher with vanishingly small error.  The second experimental result is more interesting: it shows that, when LARSA and GLRU are taught using exactly the same in-context linear regression data, they take exactly the same gradient updates.

**Strengths:**

The title of the original Transformer paper (Attention is All You Need) suggested that the Transformer is nothing more or less than a more pathlength-efficient implementation of the same set of functions that an RNN can learn.  The exact nature of the near-equivalence between Transformers and RNNs has been harder to describe than that simple first title suggested.  This paper's experimental results on the gradient update for the in-context linear regression problem are a demonstration of the closest link between Transformers and GLRUs that I have seen yet.

**Weaknesses:**

My enthusiasm is tempered by the rather extreme limitations placed on both the Transformers and the GLRUs in this paper.  Linear self-attention is far less powerful than softmax self-attention, and as demonstrated in this paper, linear gated recurrence is less powerful than nonlinear gated recurrence, so a proof of equivalence between them, while of some theoretical interest, doesn't seem to be of very high impact.

**Questions:**

Is there any reason to believe that the demonstrated equivalence would continue to hold for neural nets that include nonlinearities?

---

> ### Author Response · Authors · 2023-11-17
>
> We appreciate the reviewer’s valuable time in reviewing our paper and would like to nuance their point of view on the limitations imposed on the two architectures we consider.
>
> - As argued in the global response, there is a growing literature showing how extensions of linear self-attention layers can reach, if not surpass, Transformers-level performance. The interest of linear self-attention thus goes beyond its amenability to theoretical analysis.
> - While we provide a theoretical construction for the linear case in the main text, we show how this still holds when the gating is nonlinear in Appendix A.1. We agree with the reviewer that it makes learning more difficult, as shown in Figure 4.A. However, we think that it still helps understand the inductive bias behind architectures such as LRUs, by showing that they are more akin to reproducing linear self-attention like behaviors than other types of RNNs.
>
> We made clearer the reasons why we think our paper can have a great impact in the global response. We encourage the reviewer to consider those arguments and hope they may convince them to change their score.

---

> > ### Author Response · Authors · 2023-11-21
> >
> > The end of the discussion period is coming close. We would be most happy to discuss in case the reviewer does not find our arguments sufficiently convincing.

---

### Official Review · Reviewer_imQt · 2023-11-01

**Soundness:** 4 excellent
**Presentation:** 2 fair
**Contribution:** 4 excellent
**Rating:** 6
**Confidence:** 3

**Summary:**

The paper takes a theoretical and empirical study to relate RNNs and attention models. The authors first show a theoretical construction that simulates a single linear attention head using a gated RNN. The idea behind the construction is simple, gated RNNs accumulate key-value matrix products at each time step and use an output gated unit to compute the output using the accumulated products and the queries at each step. However, such a construction requires $O(d^4)$ parameters to simulate a $3d^2$ parameter linear attention.

Interestingly, in multiple numerical experiments to mimic linear attention, the authors still observe that such over-parametrization in gated RNNs is necessary to simulate linear attention. The authors conduct multiple structural probing experiments on trained gated RNNs to find the simulation of their construction. Furthermore, they show that existing RNN-based architectures fail to properly mimic linear attention. The authors end with interesting in-context experiments on linear regression and showcase differences in mechanisms of different RNN-based architectures.

**Strengths:**

The main strength of the paper lies in its clinical approach to connecting RNNs and attention models, which is an important question to understand for architecture design. It is an interesting approach to have a theoretical construction to understand the importance of gates in RNN models. Furthermore, the role of over-parametrization for such models has been pointed out by their theoretical construction and empirical experiments.

In addition, the in-context experiments on linear regression provide two significant observations for any future work to follow, (a) gated RNNs can simulate one step GD with even fewer neurons, and (b) other sequence-to-sequence models can perform the same task but without necessarily mimicking the behavior of one-layer attention. Thus, I believe this paper opens up interesting questions for mechanistic interpretability.

**Weaknesses:**

The main weakness of this paper lies in its slightly difficult presentation of experimental details. Here, I point out some of the difficulties that I faced when reading this paper. I additionally pose a few questions that I believe might strengthen the authors' claims.

(a) There are many experimental statements whose details aren't clear from the current version.

1. "First, we observe that only perfect memory neurons ($\lambda = 1$) and perfect forget neurons ($\lambda = 0$) influence the network output."

In Figure 2, " Only recurrent neurons with perfect memory (λ = 1, dark blue) or no memory at all (λ = 0, light grey) influence the output, consistently with the theory."

How do the authors verify this? Is this related to the pruning experiment that the authors conduct later, where they remove the neurons with any other $\lambda$ values?

2. In Figure 2, "The block structure almost perfectly matches the one of our construction". I don't understand the block structure that the authors refer to.

3. Again in Figure 2, the statement "For each output coordinate  ... which can be generated in a way that is coherent with the structure of our construction" is extremely difficult to parse.

4. In Table 2, what do the terms $x_i^j y _1$ for different $i, j$ even mean? Notations would help readers parse the results of the probing experiments.

(b)  The experiments conducted in sections 2 and 5 are with a fixed dimension. How does the loss behavior change with different parameter counts at different dimensions? Such a plot can give an empirical dependence on the order of parameters necessary with dimension.


(c) The linear regression experiments show that with sparsity in the key-value matrix, the gated RNN models can simulate more efficiently than the theoretical construction. It would be interesting to conduct similar experiments in section 2, where the authors impose low-rank/sparse constraints on the key-value matrix product and observe the empirical behavior of loss with different parameter counts.


Overall, I believe this paper will be an interesting read to the community. The current paper presentation is difficult to parse at different experimental details. Hence, I would like to interact with the authors during the rebuttal period with the questions that I posed above.

**Questions:**

Please see my questions in the previous section.

---

> ### Author Response · Authors · 2023-11-17
>
> We thank the reviewer for their positive feedback on our work and for pointing out sources of confusion. We provide some clarifications below that we hope will be helpful:
>
> (a)
>    1.  At the end of learning, not all recurrent neurons have their $\lambda$ being 0 or 1 (c.f. figure in the global answer). However, inspection of the input and output gating weights reveals that those neurons will always receive 0 as input and that they are ignored by the output gating. This is what we mean by “we observe that only perfect memory neurons (λ = 1) and perfect forget neurons (λ = 0) influence the network output”. Therefore, we can safely prune those neurons (and their corresponding input and output weights) from the network without changing its behavior. We plot the resulting network in Figure 2.A.
>    2. In our construction the input and output gating matrices have a characteristic 2x2 block structure. For example, if we look at the input gating, one of the matrix has top right and bottom left blocks that are equal to 0. We recover the same structure in $W_x^\mathrm{in}$ in Figure 2.A. The same thing holds for the 4 other weights of the gating matrix.
>    3. We will make this caption clearer. If this may help the reviewer, here is a more detailed explanation. In Figure 2.A, the output gating weights do not fully match the structure of the construction (that is one matrix with non zero values on the left side, and one with non-zero values on the right side). However, it turns out that the last 3 neurons in between output gating and the $D$ matrix (rows 11, 12 and 13 in the output gating matrices, and the corresponding columns in $D$) are functionally equivalent to one. The input weights of this neurons are described in the right part of Figure 2.B (the row and columns vectors in the right part of Fig.2.B). The nice thing is that the corresponding weights for this neuron match the expected structure: the row vector only has zero entries in the first 10 columns, and the column vector has zero entries in the last 4 rows. As a consequence, it is possible to rewrite the output gating weights (and $D$) in a way that matches the construction, while not changing at all the behavior of the network. For the mathematical details, see Appendix B.2.
>    4. Indeed, the caption was not sufficiently explaining the table. The input at time $t$ consists in the concatenated vectors $x_t=(x_{t,1},x_{t,2},x_{t,3})^\top$ and $y_t=(y_{t,1},y_{t,2},y_{t,3})^\top$. The instantaneous function for each output neuron can implement a polynomial of degree 4 in these terms. The table shows the coefficient associated to the terms of the polynomial implemented by the first neuron after training. Interestingly, the only terms without a negligible coefficients are $x_{t,1}^2y_{t,1}, x_{t,2}^2y_{t,1}$ and $x_{t,3}^2y_{t,1}$, where $x^2$ refer to $xx$. This is virtually identical to the polynomial implemented by a single step gradient descent algorithm (GD), as shown in the table.
>
> (b) In all our experiments, we vary the number of neurons and as, a consequence, the number of parameters. The reviewer might be interested in Figure 5 in the appendix, in which we look at how the loss behaves as a function of the number of parameters of the different models. We are happy to provide more information if needed.
>
> (c) We thank the reviewer for this great suggestion. We report the result of this experiment in the global answer to all reviewers.

---

> > ### Author Response · Authors · 2023-11-21
> >
> > We hope the clarifications we provided improved the presentation of our work. In any case, we remain happy to interact with the reviewer before the end of the rebuttal period.

---

### Official Review · Reviewer_2JoC · 2023-11-09

**Soundness:** 2 fair
**Presentation:** 3 good
**Contribution:** 3 good
**Rating:** 5
**Confidence:** 3

**Summary:**

The authors present a construction of a gated RNN that implements self-attention (linear) and provides a conceivable path towards RNNs that can learn self-attention. The construction relies on GLUs with a simplified rule for describing input and output gating. The authors conduct several experiments demonstrating activated neurons in the RNN correspond to scores in that would be expected in the construction. They also demonstrate parity with a linear self-attention mechanism. The authors then study features of these networks, in particular with linear regression and gradient descent, observing the impact of nonlinearity and sparsity. This work provides a theoretical foundation with which to study other approximations of self-attention.

**Strengths:**

- The construction is novel and draws a clear connection with the special case of linear self-attention.
- The explanation and construction of gated recurrent networks is clear, and the correspondence with self-attention is transparent and intuitively explained, i.e. in Figure 1.
- The idea can guide development of attention implementations with other architectures which may have implications for efficiency. Given a general foundation, future work can use similar styles of constructions to proceed.

**Weaknesses:**

- Overall, the thrust of the contribution of the paper needs to be much more clearly articulated.
  - Why is this particular construction good?
  - What is the methodology that is general enough here to use for future constructions?
  - How, explicitly, does the authors' approach pave the way for future contributions?
  - Why do the learned ideas (e.g. linear regression) strengthen the thrust of the paper.

If these ideas can be articulated more clearly in a response here and in the manuscript, I would likely change my score.

Regarding presentation:
- Worth noting that citations in the PDF version of the paper don't appear linked to citations (for me)
- Worth mentioning that GLUs in their initial construction from Dauphin et al were actually used in gated convolutional models, which resembled RNNs in their hierarchy, but were different
- While the regularization task presented in the manuscript is valuable, not having a sequence learning task holds back some of the strength of the empirical results.

**Questions:**

- Section 3.2 discusses the invertibility of the value matrix per the number of hidden neurons the RNN needs to store KVs. Under which conditions is this matrix invertible?
- In Section 4.1, how do the number of activated neurons in the construction correspond to activated attention weights? Is this correspondence clear?
- In Section 4.2, the authors describe overparameterization insofar as twice as many neurons are needed to replicate the behavior of self-attention with the RNN construction. What might the effect of regularization be here, implicit or otherwise?

---

> ### Author Response · Authors · 2023-11-17
>
> We thank the reviewer for their valuable time reviewing our paper. We have answered part of their questions in the global response and we provide additional details in the following:
>
> - **Invertibility of the value matrix.** We require the matrix $W_V$ to be invertible to decrease the number of neurons required by our construction, which of course depends on the self-attention layer we want to replicate. See the global discussion for a more comprehensive discussion on this topic.
> - **Section 4.1.** In this experiment, we observe three things when looking at the weights of the network (we provide a visualization of those weights in the global answer) at the end of learning:
>     - Most of the weights in the input gating are 0s.
>     - There are some $\lambda$ values in the recurrent that are not 0 or 1.
>     - Most of the weights in the output gating are 0s.
>
>     It turns out that recurrent neurons whose $\lambda$ is not equal to 0 or 1 have corresponding weights in the input and output gating to be 0. As a consequence, they are effectively ignored by the network. If we remove those neurons, we end up with as many neurons as in the construction. The weights of the network after this pruning are displayed in Figure 2.
>
> - **Section 4.2.** As highlighted in the previous point, not all recurrent neurons end up being used at the end of learning. However, having those extra neurons is important for learning as some neurons will get “killed” during learning. This is why overparametrization (in the sense of more recurrent neurons than the construction) is needed and we empirically observe that having twice as many neurons as in the construction is enough to reach 0 training loss. Yet, it turns out that only the neurons from the construction are actually used.

---

> > ### Author Response · Authors · 2023-11-21
> >
> > Given that the discussion period is coming to an end soon, we are gently reminding the reviewer that we are happy to engage in any discussion or provide additional clarifications.

---

### Author Response · Authors · 2023-11-17
**Global response**

We thank all the reviewers for their time and constructive feedback that will greatly help in improving the paper.

Several reviewers asked questions about the contributions of the paper, we will include more detail on why we think our result is impactful to a broad audience in the next version of the paper:

- **From an architecture design perspective.** We agree with reviewer pjc5 that the performance of vanilla linear self-attention is worse than the one of Transformers. However, LSA is the starting point of a very active area of research trying to improve it in order to match/surpass Transformers performance without suffering from its $O(T^2)$ scaling. Architectural elements inspired by deep SSMs are often integrated as in the RetNet (https://arxiv.org/abs/2307.08621), h3 (https://arxiv.org/abs/2212.14052) or, to some extent, the GateLoop architectures. Those new architectures are often performing on par, and sometimes better, than Transformers. Our construction naturally extends to them. By providing a concrete link between LSA (and extended versions) and diagonal SSMs / LRU with multiplicative interactions, we believe our result helps us understand why those new architectures work and may facilitate the design of new architectures.
- **From a mechanistic interpretability perspective.** As pointed out by reviewer imQt, we believe that our result opens interesting questions for mechanistic interpretability. For example, by showing how gated LRUs can implement linear self-attention, it becomes easier to test whether they do so in practice. In general, we believe it provides a foundation to investigate the difference in learned representations between gated LRU-like and LSA-like networks.
- **From a neuroscience perspective.** Understanding how the brain stores memory is one of the biggest open questions in neuroscience. The two prominent classes of memory models are RNNs and Hopfield networks. They are considered to behave in very different ways, RNNs store information in e.g. attractors, and Hopfield network-like memories that store associative pairs akin to LSA. Our result is thus of relevance for neuroscience: Hopfield networks can actually be implemented by RNNs. It has many interesting applications. For example, Hopfield networks are usually thought to be implemented in the brain through synaptic plasticity. This is plausible in the hippocampus, but much more unlikely in the cortex. Our results suggest the cortex may, to some extent, store associative memories in a Hopfield network-like manner, without any fast plasticity. Our construction also highlights the importance of multiplicative interactions, which are widely studied in this field.

We summarize below additional experiments ran following the reviewers’ advice:
- 3 of the reviewers asked about low-rank teachers. The $W_V$ matrix is not invertible so the compact $d(d-1)/2$ construction of Sec. 3.2 does not hold anymore. Instead, it is possible to reduce the naive construction of Sec. 3.1 to $k^2 + k$ recurrent neurons. It requires 1. applying SVD on $W_V$ and $W_K^\top W_Q$ to reduce the size of key-values and queries to $k^2$ and $k$, and 2. transfer part of $W_V$ to the output gating to $D$ to project back the original dimension. We ran an experiment in which $d=12$ but the rank of the $W_V$, $W_K$ and $W_Q$ matrices are $k=6$. We [observed](https://ibb.co/1rXpRX0) that the network finds solutions with $k^2 + k$ recurrent neurons, as predicted, and the transition characteristic of Figure 2 happens around this number.
- Reviewer 1u2u suggested running experiments on an associative recall task. In this task, the network is sequentially presented with a certain number of input-output pairs, 10 in our experiments (e.g. [d, 1, g, 3, ..., a, 9]). At the end, the model is queried with a randomly chosen seen input element and is trained to recall the corresponding output observed in context. After the initial positional dependent copy in which input-output pairs are concatenated together, which we assume to be given, an LSA layer with key size 10 is able to solve the task, as long as input-output pairs are orthogonal. We embed tokens randomly in a space of dimension 32 to achieve that. Such a LSA layer reaches a close to 0 loss in this task.
Given that the value, key, and query matrices are random matrices of size $64 \times 10$, we can consider that they are of rank 10. Therefore, as mentioned in the previous point, there exists a construction of size $10^2 + 10 = 110$ that reach 0 loss. However, we [observe](https://ibb.co/yyzjVHF) the typical transition other tasks exhibit before that number (around 35), suggesting that the RNN has learned a more compact solution. We will continue investigating this promising direction and provide the results in the next version of the paper.
- Reviewers 2JoC and imQt asked to clarify the pruning procedure. We provide a visualization of the non-pruned weights [here](https://ibb.co/FBmgLRj).

---

### Meta-Review · Area_Chair_2Dmr · 2023-12-13

**Metareview:**

This paper investigates the relationship between gated recurrent layers and linear self-attention. It demonstrates that a gated recurrent layer, when properly parameterized and combined with a linear readout, can emulate any linear self-attention layer. While these findings are novel and interesting, the reviewers have expressed concerns about the limited impact of the work, primarily because it only focuses on the linear setting. Additionally, significant revisions are necessary to enhance the paper's clarity and presentation quality. In its current state, the paper is not prepared for publication. Another round of revision could greatly benefit this work.

**Justification For Why Not Higher Score:**

This is overall okay paper, and the findings are novel and interesting. However, the impact is limited. The paper has potential to stimulate more research in this area, which would justify a higher sore. However, it is not clear if the ideas can be extended to non-linear self-attention. I strongly feel that the paper can benefit a lot from another round of iteration before publication.

**Justification For Why Not Lower Score:**

N/A

---

### Decision · Program_Chairs · 2024-01-16

Reject